# Description and Evaluation of Dye and Contrast Media Distribution of Ultrasound-Guided Rectus Sheath Block in Cat Cadavers

**DOI:** 10.3390/ani14121743

**Published:** 2024-06-09

**Authors:** Gonzalo Polo-Paredes, Marta Soler, Francisco Gil, Francisco G. Laredo, Amalia Agut, Sara Carrillo-Flores, Eliseo Belda

**Affiliations:** 1Departamento de Medicina y Cirugía Animal, Facultad de Veterinaria, Universidad de Murcia, 30100 Murcia, Spain; gpolo@um.es (G.P.-P.); mtasoler@um.es (M.S.); laredo@um.es (F.G.L.); amalia@um.es (A.A.); 2Hospital Veterinario Universidad de Murcia, 30100 Murcia, Spain; sacarriflores@gmail.com; 3Escuela Internacional de Doctorado de la Universidad de Murcia, Programa en Ciencias Veterinarias, Universidad de Murcia, 30100 Murcia, Spain; 4Departamento de Anatomía y Anatomía Patológica Comparada, Facultad de Veterinaria, Universidad de Murcia, 30100 Murcia, Spain; cano@um.es

**Keywords:** abdominal analgesia, fascial block, feline, locoregional anaesthesia, rectus sheath block, ultrasound-guided

## Abstract

**Simple Summary:**

The rectus sheath block is an ultrasound-guided locoregional anaesthetic technique which aims to provide analgesia to the midline of the abdomen. Previous studies in dog cadavers have shown that the injection of methylene blue within the rectus sheath stained the rami ventrales from T10 to L1, potentially providing anaesthesia to the cranial and middle abdominal midline. We hypothesise that a similar approach would be feasible in cats and could offer equivalent results to those obtained in dogs. For this purpose, ten cat cadavers were included. An iopromide–methylene blue mixture (0.4 mL kg^−1^) was injected between the rectus abdominis muscle and the rectus sheath. Then, a computed tomography scan and anatomical dissection were carried out to assess the injectate spread. Our results showed a consistent staining of the rami ventrales of several spinal nerves (T11–T12) and frequent staining of T10 and T13. These results are compatible with the supply of somatic anaesthesia in the cranial abdominal midline, although the middle and caudal midline anaesthesia could not be provided.

**Abstract:**

The rectus sheath block is an ultrasound-guided anaesthetic technique which aims to provide analgesia to the abdominal midline. This study aimed to assess the distribution of 0.4 mL kg^−1^ of a mixture of methylene blue and iopromide injected into each hemiabdomen in the internal rectus sheath in cat cadavers. We hypothesise that this technique would be feasible and would cover the rami ventrales of the last thoracic and the first lumbar spinal nerves. The study was divided into two phases. Phase 1 aimed to study the anatomical structures of the ventral abdominal wall (four cats were dissected). Phase 2 (ten cadavers) consisted of an ultrasound-guided injection of the mixture mentioned above and the assessment of its distribution by computed tomography and anatomical dissection. The results showed the staining of the cranioventral abdominal wall with a craniocaudal spread of four (three to eight) vertebral bodies. Methylene blue stained three (one to four) rami ventrales, affecting T10 (60%), T11 (100%), T12 (90%), T13 (50%) and L1 (5%). Based on these results, it could be stated that this technique could supply anaesthesia to the midline of the abdominal midline cranial to the umbilicus in clinical patients, but it may not be able to provide anaesthesia to the middle and caudal midline abdominal region.

## 1. Introduction

Fascial ultrasound-guided blocks are specialised techniques which aim to desensitise extensive anatomical areas thanks to the locoregional anaesthetic distribution allowed by superficial fascial planes [1,2,3]. The inclusion of these techniques into balanced anaesthetic protocols improves patient comfort, reduces perioperative opioid consumption and enhances recovery after surgery [4]. Recently, new ultrasound-guided anaesthetic techniques such as the transversus abdominis plane (TAP) [5,6] and the quadratus lumborum (QL) [7,8,9] blocks have been described in cats in both cadaveric and clinical studies [10,11]. The objective of these techniques is to provide analgesia to the abdomen. In the same line, the rectus abdominis sheath (RS) block [12] has been proposed to provide anaesthesia to the abdominal midline in people [13,14,15,16,17]. The rectus abdominis internal sheath is formed by the aponeurosis of the transversus abdominis muscle and, cranial to the umbilicus, also by the aponeurosis of the obliquus internus abdominis (OIA) [18,19]. The RS block [20] consists of the injection cranial to the umbilicus of a high volume of a local anaesthetic between the rectus abdominis (RA) muscle and its internal sheath [19,21].

In veterinary medicine, cadaveric studies of the ultrasound-guided RS block have been published in dogs [22], horses [23,24], calves [25] and pigs [26]. Their results showed the consistent staining of the nerves responsible for the cranial and middle abdominal midline [27] innervation. However, they were not able to show involvement of the caudal midline, as no RS is found in the caudal abdominal region. Additionally, clinical studies have also been conducted in dogs [28], equines [23] and calves [29] showing results compatible with the cadaveric studies. New clinical studies have also been conducted in cats [30,31].

To the authors’ knowledge, no studies have investigated the distribution and nerve staining of the RS block in feline cadavers. We hypothesise that this technique would cover the rami ventrales (RVs) of the last thoracic spinal nerves and the more cranial lumbar spinal nerves. We aimed to assess the distribution of a 0.4 mL kg^−1^ mixture of dye and contrast media, ultrasound-guided administered between the rectus abdominis muscle and its internal sheath RS block in cats.

## 2. Materials and Methods

This research obtained approval from the University of Murcia’s Biosecurity Committee in Experimentation (CBE 553/2023). Throughout the study, 17 cat cadavers were assessed for eligibility, dead or euthanised for reasons unrelated to the current study. Cadavers were donated to the University of Murcia voluntarily by their owners through the Donation Program of the Veterinary Faculty (PDCAVetMu). Once the cat cadavers were donated, the bodies were promptly frozen after death. Forty-eight hours before use, the bodies were thawed at a temperature between 20 and 22 °C.

Inclusion criteria covered cat cadavers whose body condition score (BCS) was superior to 2/9 and inferior to 8/9 according to the WSAVA Classification [32], and no trauma or anatomical alterations were found in the abdominal region.

The study was then divided into the following two phases.

### 2.1. Phase 1: Anatomical Study

In this phase, we aimed to assess the anatomical features of the ventral abdominal region from the processus xiphoideus to the tuberculum pubicum. Four cat cadavers were used.

The thorax and abdomen walls were clipped, and the cadavers were positioned in dorsal recumbency. After performing an incision on the midline, the pectoralis profundus (PP), pectoralis transversus (PT), obliquus externus abdominis (OEA), OIA, TA and RA were dissected. The relation between these muscles was assessed. Then, the RVs of the nerves from the ninth thoracic (T9) to the third lumbar (L3) spinal nerves were identified in the lateral wall, and their path until reaching the midline was assessed. Finally, an incision on the linea alba was carried out and the nerves were followed until its origin. G.P.-P. and F.G. performed this phase.

### 2.2. Phase 2

#### 2.2.1. Ultrasound-Guided Technique

Thirteen cat cadavers were assessed for eligibility. Two syringes of 0.4 mL kg^−1^ each of a mixture of 50:50 iopromide (300 mg mL^−1^, UltraVist300, Bayer, Berlin, Germany) and methylene blue (10 mg mL^−1^, Pancreac Quimica, AppliChem, Castellar del Vallès, Spain) were used.

After clipping the bodies, as mentioned above, and positioned in dorsal recumbency, a 3–13 Hz linear ultrasound probe (SL1543, MyLabGamma Vet, Esaote, Florence, Italy) was positioned in the midline, transversal to the median plane, 2 cm cranial to the umbilicus. The linea alba and RA were identified, and the depth and gain were configured to optimise the image quality.

Then, the probe was glided laterally until the OEA and the TA were identified. At this point, a sonovisible needle (Ultraplex 20 G 0.9 × 100 mm 30°, BBraun, Melsungen, Germany) was advanced in plane in a mediolateral direction towards the conjunction of the RA and the TA (Figure 1).

Once the external sheath of the RA was pierced, the needle was reoriented and advanced through the RA until being in contact with the RS at the medial edge of the belly of the TA (Figure 2). At this location a volume test of 0.1 mL of injectate was administered. If the image was compatible with the expected distribution (the anechoic pocket of injectate between the RA and its RS), the whole volume (0.4 mL kg^−1^) was injected. The technique was repeated in the opposite abdomen. The needle visualisation throughout the process was registered. All of the ultrasound-guided injections were made by the same researcher (G.P.-P.).

#### 2.2.2. Computed Tomography (CT) Study

Once the ultrasound technique was carried out (10–20 min), the cadavers were subjected to CT scans (dual-slice CT scanner, General Electric HiSpeed, General Electric Healthcare, Madrid, Spain) of the region between T9 and the pubis. The bodies were placed in dorsal recumbency with their upper and lower extremities extended. The collimator pitch was set to 1, the slide thickness was set to 3 mm, the reconstruction interval was set with a 50% overlap, the kVp was set to 120 and the mA was set to 100. Standard bone and soft tissue reconstruction algorithms were employed. The reformatted images were evaluated by two radiology experts (M.S. and A.A.) who assessed the location and distribution of the contrast solution. The craniocaudal spread of the contrast media was expressed in number of vertebral bodies (VBs) it parallelly covered.

#### 2.2.3. Dyeing Study

Immediately after the CT scan, the bodies were dissected. A ventral midline incision was carried out and the skin was dissected up to the dorsal region to expose the underlying tissues. After removing the PP and PT muscles from their origin at the sternebrae, the OEA muscle was dissected from its origin at the caudal border of the ribs until reaching the RA. The RS was dissected, and the number and location of the nerves stained were assessed. A nerve was considered positively stained when a minimum of 1 cm of its length was dyed in all its circumference [5,6]. G.P.-P. and F.G. performed all the dissections.

### 2.3. Statistical Analysis

A statistical descriptive test was carried out using Microsoft Excel 365 software (Microsoft Corporation, Redmont, WA, USA) and Real Statistics Resource Pack plug-in software (release 7.6, Copyright 2013–2021, Charles Zaiontz, www.real-statistics.com accessed on 16 December 2023). The Shapiro–Wilk test was used to evaluate the distribution of the data. The results were expressed as mean ± standard deviation (SD) if normally distributed, or median (range) if non-normal distribution was found.

## 3. Results

### 3.1. Phase 1: Anatomical Study

One male and three female cat cadavers underwent anatomical dissection: three European shorthairs and one Persian. They weighed 3.7 (1.5–3.7) kg and all of them were scored four out of nine on the BCS.

The thoracic aspect of the OEA was covered ventrally by the PP and PT. Three different muscle patterns were found in the abdomen of the cat cadavers (Figure 3). In the cranial abdomen, the RA was found surrounded ventrally by the OEA muscle, dorsoventrally by the aponeurosis of OIA and dorsally by the TA muscle. In the middle abdominal region, the RA was covered ventrally by the OEA and the aponeurosis of OIA, and dorsally by the aponeurosis of TA muscle. In the caudal region, the TA muscle shifted to a ventral position in relation to the RA, causing a discontinuation of the RS. Thus, in the cranial abdomen, the RS is formed by the OIA and TA aponeurosis, while, in the middle abdomen, only by the TA aponeurosis. Finally, the RS ended in the aforementioned change in the TA position in the caudal region.

The RVs of the thoracic nerves from T9 to the twelfth thoracic nerve (T12) were observed caudal to their respective ribs at the lateral wall of the thorax. The ramus ventralis of T9 ended at the lateral aspect of the processus xiphoideus. The tenth (T10), eleventh (T11) and twelfth (T12) RVs were found emerging from the costal arch and advancing between the OIA aponeurosis and the TA towards the RA muscle, where they ended, cranial to the umbilicus. Finally, at the lateral aspect of the abdominal wall, the RVs of the thirteenth (T13), first (L1), second (L2) and third (L3) lumbar nerves were also found lying between the OIA and TA muscles. At this point, the RV of T13 continued until it ended in the RA caudal to the umbilicus. However, the RV of L1 followed the same path as the TA, but it slid through the interface where the TA changed from a dorsal to a ventral position with regard to the RA muscle. In addition, the RV of L2 ended at the linea alba, ventral to the RA muscle, while the RV of L3 ran caudally to innervate the inguinal area. Both nerves were not included in the RS compartment. Thus, the RS embraced the RVs from T10 to T13 or L1.

### 3.2. Phase 2

#### 3.2.1. Demographic Distribution

Thirteen cadavers were assessed for eligibility in this phase. However, three of them were excluded due to a tear in the muscular abdominal wall, a mammary mass and extreme cachexia observed after thawing. Finally, ten cadavers (20 hemiabdomens) were included in this phase; four males and six females. They weighed 2.7 (1.6–6) kg and had a BSC of four (three to seven) out of nine).

#### 3.2.2. Ultrasound-Guided Technique

The three muscles involved (RA, TA and OEA) were identified in all (20/20) hemiabdomens (Figure 2) The visualisation of the needle was clear and well defined along its path in all injections performed (20/20).

#### 3.2.3. Computed Tomography Study

The spread of the contrast media was observed in the target area in all hemiabdomens (20/20). The maximal craniocaudal spread was four (three to eight) VBs (Figure 4). A small amount of contrast media was observed intramuscularly into the RAs in all the hemiabdomens. Contrast media was also found in the TAP in 3 hemiabdomens (3/20). In one cadaver, contrast media was also found in the peritoneal cavity (1/10). Finally, in the cat with the highest weight (6.0 kg) and BCS (seven out of nine), contrast media was found spreading from the arcus costalis to the pelvis.

#### 3.2.4. Spread Study

Methylene blue was found in the target area in all hemiabdomens (20/20) (Figure 5), and in 3 of them (3/20), dye was also found in the TAP. In 1 cat cadaver (1/10), colorant was observed in the peritoneal cavity. Methylene blue stained three (one to four) RVs, affecting T10 (60%), T11 (100%), T12 (90%) and T13 (50%) (Figure 6). L1 was stained in one hemiabdomen (5%).

## 4. Discussion

Our results state that an ultrasound-guided injection of 0.4 mL kg^−1^ of a mixture of methylene blue and iopromide can consistently stain the RVs of T12 and T13 and frequently stain T10 and T13. These findings are therefore compatible with the induction of analgesia in the cranial abdomen and potentially in the middle abdomen.

Our anatomical findings in cat cadavers showed that the ventral abdominal wall of cats is innervated by the RVs of the spinal nerves from T10 to L3 [5,6,18]. The ramus ventralis of T9 ends at the xiphoid cartilage and it does not seem to play an important role in most of the abdominal procedures. In addition, in the caudal abdomen, the RS is discontinued as the TA changes from dorsal to ventral with respect to the RA. At this point, the ramus ventralis of L1 follows a similar path to that of the TA, so its integration into the RS remains unclear. However, the RVs of L2 and L3 run caudal to the change in the TA position, where the RS is discontinued. These anatomical results are in agreement with those previously reported by Garbin et al. (2022) [6] and could explain why these nerves were not stained in the present study.

Clinical research with human volunteers showed that the TAP block may not be able to desensitise the midline, thus not providing anaesthesia to an incision in the linea alba [33,34]. Those authors reported a patchy block without midline involvement. However, the reason why these results were obtained is not clearly stated. Considering these studies, the RS block could be an advantageous locoregional technique over the TAP block regarding midline surgeries in the cranial and middle abdomen. Nevertheless, clinical studies in both human [19] and veterinary medicine [10,35] have also shown the analgesic properties of the TAP block over the ventral wall of the abdomen. Further clinical studies are needed to clarify which would be the most appropriate locoregional technique to provide analgesia to the ventral abdominal wall.

Our approach to the RS was different from that previously described by St James et al. (2020) in canine cadavers [22]. The external RA sheath of cats is a hard and stiff connective tissue layer. The pressure exerted to pierce this external sheath increases the risk of perforating the entire thickness of the RA, causing an accidental intraperitoneal or visceral puncture. That is the reason why we decided to modify the St James et al. approach by introducing the needle in a mediolateral direction, given the presence of the last portion of the TA muscle under the lateral third of the RA. Although the TA is a thin muscle, it can offer an additional protective barrier to prevent peritoneal perforation. The presence of the OIA aponeurosis (Figure 3) in this location ensures injection within the RS. Furthermore, in the authors’ experience, this modified technique adapted better to the anatomy of the cat cadavers, making an “in plane” approach easy in small individuals. However, further comparative research should be conducted to clarify if a mediolateral approach truly decreases the aforementioned risk and increases the ease of performing the spread of the injectate and nerve staining.

The iopromide distribution observed on the CT images was similar in all the cadavers but one, which had the highest BCS and weight. In this individual, the contrast media was distributed from the arcus costalis to the pelvis. A higher total volume, when compared with the rest of the cadavers, could have been the reason for this finding, since the volume was calculated based on the real, not ideal, body weight. The mixture (one-to-one) employed was chosen to make the results comparable to other studies and block techniques in dogs [36,37,38] and cats [9]. Other solution concentrations have been studied in cats [6], showing that lower contrast concentrations could be employed. In addition, one study in dogs showed differences in the spread extension depending on the contrast concentration employed [39]. The total 0.8 mL kg^−1^ volume was chosen to avoid exceeding, in a clinical scenario, the maximum recommended doses (2 mg kg^−1^) of some of the most commonly used local anaesthetics in cats (bupivacaine 0.25% and ropivacaine 0.25%) [40,41]. A further dilution of the local anaesthetic (bupivacaine 0.125% or ropivacaine 0.125%) could increase the total volume without exceeding the maximum recommended dose. However, a reduction in the concentration of the drug could reduce its effectiveness [42]. Furthermore, a recent study developed in cats [43] stated that higher doses of bupivacaine 0.25% (1 mL kg^−1^) can be safely administered in TAP blocks. Perhaps a volume of 1 mL kg^−1^ could increase the caudal distribution of the injectate when an RS block develops, increasing the percentage of the staining of the ramus ventralis of L1.

Due to the difference in the size and weight of our cat cadavers, it was decided to express the maximal craniocaudal iopromide spread in a number of VBs. The authors thought this measure could have more clinical relevance and could be compared to other studies which use this same scale with different anaesthetic techniques [9,38].

As a consequence of the small size of the cats, the size and orientation of the needle bevel could play an important role in the distribution of the injectate when performing fascial plane blocks in this species [6,9]. These authors reported the use of different needle sizes and bevel angles (22 G, 20° for the TAP block study and 20G, 30° for the QL). In the current study, as previously described, a blunt bevel (30°) was used. The intraperitoneal and TAP distribution of the contrast media was found in one and three cadavers, respectively, probably due to inadvertent punctions of the RS and peritoneum. Unwanted punctions of the peritoneum were also described by Garbin et al. (2022) [6] when performing TAP blocks in cat cadavers. These facts probably correlate the size and orientation of the bevel with the thinness of the RS due to the size of the cat cadavers. Intramuscular distribution of the contrast media was also observed in all of the RA muscles, which could contribute to a slight reduction in the spread of the injectate. The type of needle employed in our study (20 G) could also contribute to the distribution of the injectate. A sonovisible needle was chosen in this study to enhance its visualization throughout its path [44]. The use of smaller-diameter needles could have decreased the intramuscular or TAP distribution of the injectate observed in some of the cats. However, considering the size of the fascias in this species, even smaller needle diameters could not completely rule out this fact [3].

The nerves stained using this RS block approach in the present study ranged from T10 to T13. In a clinical scenario, this technique could potentially provide anaesthesia cranial to the umbilicus and, potentially, the middle abdominal midline. Compared to a previous study conducted on cadaver Beagle dogs using a higher volume of injectate (0.5 mL kg^−1^) and a different injection point (at the level of the umbilicus) [22], we had the same nerves stained in our study. However, the percentage of nerve staining was slightly different, showing our results had a more consistent cranial distribution of the dye, potentially due to the more cranial injection point.

This locoregional block has shown inhalant anaesthetic-sparing properties [31]. However, little or none of the analgesic properties have been observed in cats undergoing ovariectomy and ovariohysterectomy [30,31]. At this point, it must be considered that the usual incision points in these surgeries are caudal to the umbilicus, being the area innervated by the RVs of T13, L1 and L2. To the authors’ knowledge, our study is the first one showing the distribution of the RS block in cat cadavers, and based on the dye distribution we observed, this technique may not provide full analgesia to this surgical approach. However, it may be appropriate for procedures with a midline incision cranial to the umbilicus. Further studies approaching the RS block caudally to the umbilicus are needed to elucidate whether this modification could increase the percentage of the staining of the ramus ventralis of L1. Another important consideration is that, as with the TAP block, the RS block is only providing somatic analgesia to the abdominal wall. This fact must be taken into account in surgeries with an intense visceral manipulation, in which different locoregional anaesthetic techniques (epidural, QL or caudal paravertebral blocks) must be considered.

The analgesic properties on the abdominal wall of different ultrasound-guided blocks have been studied in both people and animals [19,45]. Among them, one of the most traditionally described is the TAP block [46,47]. This technique has been largely studied in cat cadavers [5,6], and also in clinical cases in cats undergoing ovariohysterectomy [10], showing opioid-sparing properties. On the same line, a more recently described ultrasound-guided block is the QL [48,49]. Within the last few years, this block has been studied in cat cadavers [7,8,9], showing a nerve stain distribution compatible with analgesia in the middle and caudal abdomen, as well as visceral abdominal analgesia. However, to the authors’ knowledge, no comparison of the QL and the RS blocks has been studied in feline patients, and just one recent report compared the TAP and the RS blocks in dogs [28]. In addition, combinations of two or more of these techniques could enhance local anaesthesia properties and its area of desensitization. Further clinical studies are needed to assess which locoregional techniques or combinations would provide better analgesia as well as anaesthetic and opioid-sparing effects.

Several limitations can be found. This study involved only a few cadavers, which could result in a bias. The ultrasound images could differ in a clinical scenario due to changes from freezing and thawing. Likewise, differences in the hydration, muscle tension and the physicochemical properties of the mixture of iopromide and methylene blue compared to the commonly used local anaesthetics could have altered its distribution pattern. The 1 cm length nerve staining was chosen, as it is considered enough to block the conduction of the action potential of the neuron [50]. However, the local anaesthetic concentration, the use of adjuvants and the tissue pH could have modified this fact [51]. Furthermore, the abdominal wall movement during respiration could promote further spreading in live cats. It should also be considered that no large or small cadavers, or bodies with a BCS over seven out of nine or under three out of nine were involved. This aspect could influence the injectate distribution due to changes in muscle tension and fat deposits. Finally, bias could be found considering that the same researcher performed both the injections and the dissections.

## 5. Conclusions

This study showed the consistent staining of the T11 and T12 RVs and the frequent staining of the T10 and T13 RVs. A spread of injectate caudal to L1 was not observed and the staining of this nerve was anecdotical. Therefore, the RS block would be a suitable option to provide analgesia in live cats undergoing midline laparotomies cranial to the umbilicus, and it potentially may also cover the middle abdominal midline, but not the caudal abdomen. Further studies are necessary to improve the technique or investigate the combination of the RS block with the lateral TAP block to desensitize all nerves responsible for the innervation of the abdomen.

## Figures and Tables

**Figure 1 animals-14-01743-f001:**
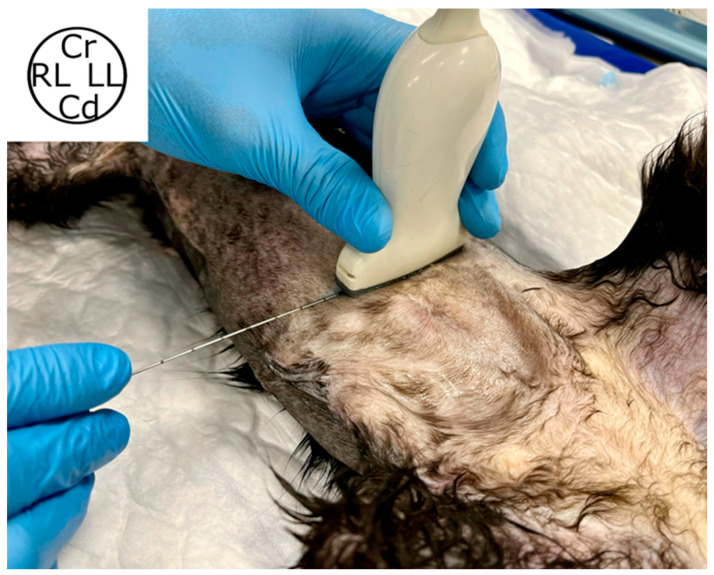
Transducer position during RS block ultrasound-guided injection. The cat cadaver was positioned in dorsal recumbency, the probe was placed transversal to the midline, 2 cm cranial to the umbilicus. The needle was advanced “in plane”. Cr, cranial; Cd, caudal; RL, right lateral; LL, left lateral.

**Figure 2 animals-14-01743-f002:**
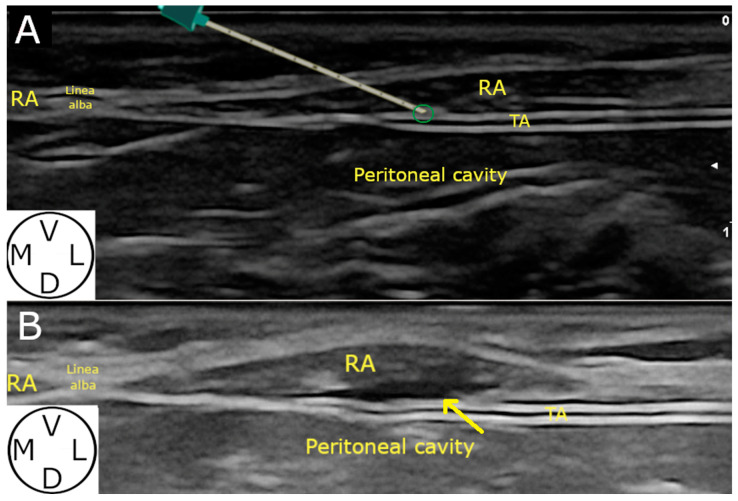
Ultrasound image of the rectus sheath block in the cat cadavers. (**A**) The needle is introduced “in plane” and advanced in a mediolateral direction through the RA until reaching the rectus internal sheath (green circle), characterized by a hyperechoic line under the muscle. (**B**) Anechoic pocket (yellow arrow) observed after injection of the injectate. TA, transversus abdominis muscle; RA, rectus abdominis muscle; M, medial; L, lateral; V, ventral; D, dorsal.

**Figure 3 animals-14-01743-f003:**
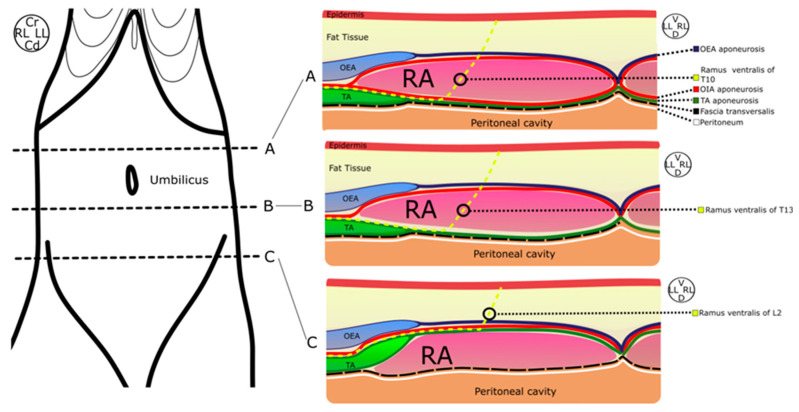
Schematic illustration of the anatomic structures surrounding the rectus abdominis muscle throughout the abdomen. (A) Transversal view of the cranial abdomen. (B) Transversal view of the middle abdomen. (C) Transversal view of the caudal abdomen. Cr, cranial; Cd, caudal; LL, left lateral; RL, right lateral; V, ventral; D, dorsal; OEA, obliquus externus abdominis muscle; OIA, obliquus internus abdominis muscle; TA, transversus abdominis muscle; RA, rectus abdominis muscle.

**Figure 4 animals-14-01743-f004:**
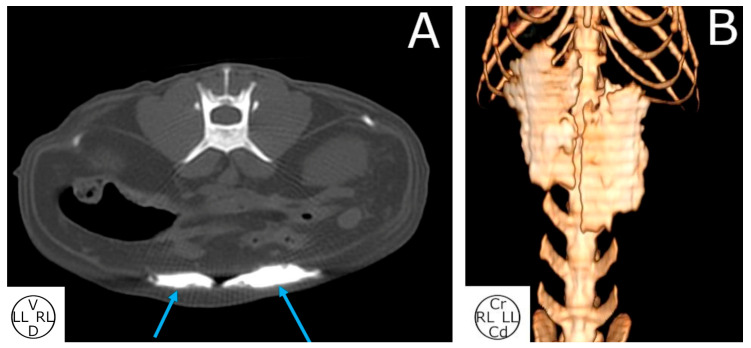
Computed tomography transverse image at the level of the third lumbar vertebra with bone window (**A**) and 3D-VR reconstructions (**B**) after the bilateral administration of 0.4 mL kg^−1^ of a mixture of methylene blue and iopromide into the RS, 2 cm cranial to the umbilicus of the cat cadavers. (**A**) Contrast can be observed in the RS (blue arrows). (**B**) Volume-rendered 3D reconstruction of the area between the ninth thoracic and fifth lumbar vertebrae. RS, rectus abdominis internal sheath; RA, rectus abdominis muscle; D, dorsal; LL, left lateral; RL, right lateral; V, ventral; Cr, cranial; Cd, caudal.

**Figure 5 animals-14-01743-f005:**
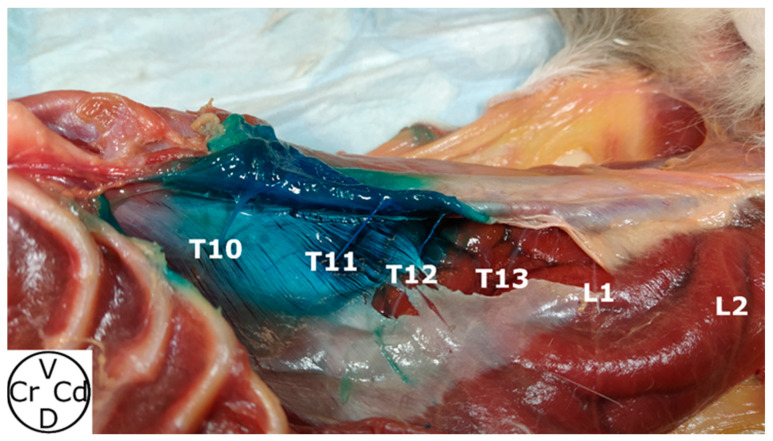
Anatomical dissection of the ventral abdominal area dyed after methylene blue injection where the rami ventrales of T10, T11, T12, T13, L1 and L2 can be observed. In this hemiabdomen, the rami ventrales of T10, T11 and T12 were stained. The T10, T11, T12, T13, L1, L2 and L3 rami ventrales of T10, T11, T12, T13, L1 and L2 spinal nerves, respectively; Cr, cranial; Cd, caudal; V, ventral; D, dorsal.

**Figure 6 animals-14-01743-f006:**
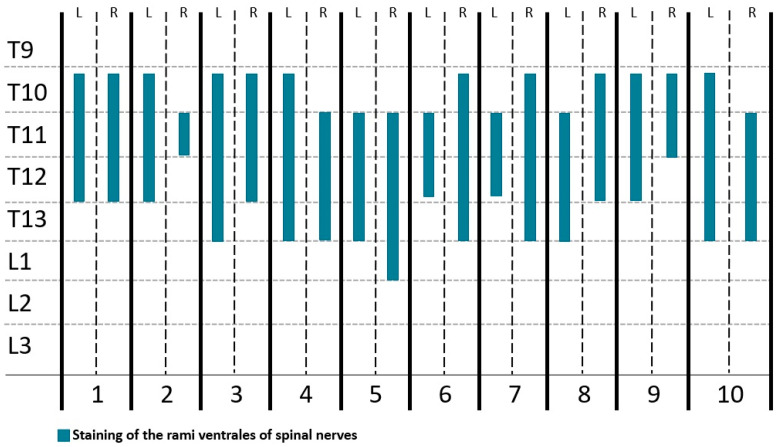
Number of RVs from T9 to L3 stained after the injection of 0.4 mL kg^−1^ of methylene blue and iopromide by an RS block technique in 10 cat cadavers (numbers 1–10). RS, rectus abdominis internal sheath; L, left hemiabdomen; R, right hemiabdomen; 1–10, cats 1–10; T9, T10, T11, T12, T13, L1, L2 and L3 rami ventrales of T9, T10, T11, T12, T13, L1, L2 and L3 spinal nerves, respectively.

## Data Availability

Data supporting the reported results can be sent to anyone interested by contacting the corresponding author.

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
