# Peer review of "Description and Evaluation of Dye and Contrast Media Distribution of Ultrasound-Guided Rectus Sheath Block in Cat Cadavers"

_animals, 2024, doi:10.3390/ani14121743_

Round 1

Reviewer 1 Report

Comments and Suggestions for Authors

Thank you to the authors for this excellent work. It is very well-organized, illustrative, and well-designed. The topic is important. I find it interesting that animals were used without any blockade to understand the species' anatomy before performing ultrasound or administrations.

The main topic I would like to see clarified is whether the consistently stained nerves are responsible for the innervation of the middle abdominal wall or only the cranial part.

Throughout the document, some key references are missing to justify certain statements.

Introduction:

Line 49 - Please use veterinary references whenever they exist, as is the case here. It does not seem necessary to use human references.

Materials and Methods:

Line 119 - A volume of 0.1 ml seems a bit small to confirm the application site. It was clearly visible that it was located between RA and RS.

Figure 2 - I suggest adding an image after the inoculation.

Line 130 - How long was the wait between administration and performing the CT? Please add the exact time in the document.

Line 147 - What was the reason for using 1 cm? If it was based on any literature, please indicate which.

Statistical Analysis:

  • How was the sample size calculated? Please provide information regarding the sample power test.

Results:

Line 160 - I suggest including the mean value and standard deviation. It is easier for the reader.

Line 194 - Use mean value + standard deviation.

Table 1 - I do not consider it essential. I suggest removing the table, as the information is already described earlier in the text.

Figure 6 - Please improve the image quality.

Were the animals that had less dispersion, such as 2 right, the ones where intraperitoneal contrast was observed?

Discussion:

Lines 235-237 - Considering that the contrast was consistently present up to T13, can it be stated that there will be analgesia in the middle abdomen? Please use a reference.

Lines 275-279 - This is a bit confusing. It states that the dose used is the most common but not necessarily safe, and that only study 35 confirms this dose is safe. Please rewrite these lines. Provide a reference for the recommended dose of 2 mg/kg for bupivacaine or ropivacaine in cats. The total dose in study 35 is 2.5 mg/kg, not 1 mg/kg as described.

Lines 280-281 - Is there any study that uses this volume? Why is it mentioned? Please provide a reference.

Lines 291-295 - “As a consequence of the small size of the cats, the size and orientation of the needle 291

bevel could play an important role in the distribution of the injectate when performing 292

fascial plane blocks in this species [6,9]. These authors reported the use of different needle 293

sizes and bevel angles (22G and 20G, 20º and 30º respectively). In the current study, as 294

previously described, a blunt bevel (30º) was used.” - I suggest placing this description after the ultrasound description - that is, in line 270.

Line 309 - Please indicate the source for the statement “This locoregional block has shown inhalant anaesthetic-sparing properties.”

Lines 311-315 - I agree. Here it is stated that it can only desensitize the cranial part, but the rest of the document mentions the cranial and middle abdominal wall. Please clarify.

Lines 328-329 - I suggest removing the sentence “On the same line, a more recently described ultrasound-guided block is the QL [41,42]” and references. They do not add anything to the work.

Line 333 - Please remove the author's name and include just the number according to the journal's formatting rules.

Conclusion:

Clarify the issue of the middle abdomen.

Author Response

Introduction:

Q: Line 49 - Please use veterinary references whenever they exist, as is the case here. It does not seem necessary to use human references.

A: We have included Campoy et al. as a reference for fascial blocks in veterinary patients.

Materials and Methods:

Q: Line 119 - A volume of 0.1 ml seems a bit small to confirm the application site. It was clearly visible that it was located between RA and RS.
A: We agree. However, given that total volumes had a range between 0.63 mL and 2.4 mL, using a higher volume to confirm the location would have resulted in a volume waste that could have altered the distribution and nerve staining.

Q: Figure 2 - I suggest adding an image after the inoculation.

A: A second subfigure (B) has been added showing the anechoic pocked formed after the mixture injection.

Q: Line 130 - How long was the wait between administration and performing the CT? Please add the exact time in the document.

A: Information included

Q: Line 147 - What was the reason for using 1 cm? If it was based on any literature, please indicate which.

A: This information is now included in the Limitations. And two new references were added:

-  Raymond, S.A.; Steffensen, S.C.; Gugino, L.D.; Strichartz, G.R. The Role of Length of Nerve Exposed to Local Anesthetics in Impulse Blocking Action. Anesth Analg 1989, 68, 563–570.

- Otero, P.E.; Portela, D.A. Manual of Small Animal Regional Anesthesia: Illustrated Anatomy for Nerve Stimulation and Ultrasound-Guided Nerve Blocks; Fuensalida, S.E., Romano, M., Eds.; Second edition.; Editorial Inter-Medica: Buenos Aires, 2019; ISBN 978-950-555-465-2.

Q: Statistical Analysis:

  • How was the sample size calculated? Please provide information regarding the sample power test.

A: Being this a cadaveric study sample size was not calculated as we try to reduce the number of cadavers needed to the minimum (3R). Sample size between 6-10 cadavers are commonly accepted as adequate in this kind of studies.

This is a similar practice than in many other cadaveric studies:

Alaman et al. https://doi.org/10.1016/j.vaa.2021.09.002  

Viscasillas et al. https://doi.org/10.3390/ani11102945"https://doi.org/10.3390/ani11102945

Garbin et al. https://doi.org/10.1016/j.vaa.2020.08.003"https://doi.org/10.1016/j.vaa.2020.08.003

Marchina-Gonçalves et al. https://doi.org/10.3390/ani12010018"https://doi.org/10.3390/ani12010018

, etc.  

Results:

Q: Line 160 - I suggest including the mean value and standard deviation. It is easier for the reader.

A: Thank you for your suggestion, median (range) has been added, as these 4 cats did not follow normal distribution, probably due to the small size used in the anatomical study.

Q: Line 194 - Use mean value + standard deviation.

A: As detailed in the section “2.3. Statistical analysis”, the values that did not follow normal distribution are expressed in “median (range)”. This is the case for the weight and the BCS.

Q: Table 1 - I do not consider it essential. I suggest removing the table, as the information is already described earlier in the text.

A: We appreciate your suggestion, table 1 has been deleted

Q: Figure 6 - Please improve the image quality.

A: The image quality and size have been increased.

Q: Were the animals that had less dispersion, such as 2 right, the ones where intraperitoneal contrast was observed?

A: No, this cat was one of the cats in which TA dispersion was found. The one with intraperitoneal contrast was 7 left (2 RV stained- T11 and T12).

Discussion:

Q: Lines 235-237 - Considering that the contrast was consistently present up to T13, can it be stated that there will be analgesia in the middle abdomen? Please use a reference.

A: Thank you for your comment, this statement has been one of the most debated points between the authors, since T13 staining was frequent (50%) but not consistent.

These sentences have been rephrased

Q: Lines 275-279 - This is a bit confusing. It states that the dose used is the most common but not necessarily safe, and that only study 35 confirms this dose is safe. Please rewrite these lines. Provide a reference for the recommended dose of 2 mg/kg for bupivacaine or ropivacaine in cats. The total dose in study 35 is 2.5 mg/kg, not 1 mg/kg as described.

A: Both the references already included in the text (Monteiro’s and Grubb’s) include the maximum recommended doses for ropivacaine and bupivacaine (which differ slightly between both authors)

The different units of measure can be misled. The dose used in the study mentioned is certainly 2.5 mg/kg, but it is stated that a volume of 1 milliliter (not milligram) per kilo of bupivacaine 0.25% is safe.

Q: Lines 280-281 - Is there any study that uses this volume? Why is it mentioned? Please provide a reference.

A: This volume (1 mL Kg-1) is the one utilized in the study referenced at the beginning of the sentence (Garbin 2022)

Q: Lines 291-295 - “As a consequence of the small size of the cats, the size and orientation of the needle 291 bevel could play an important role in the distribution of the injectate when performing 292 fascial plane blocks in this species [6,9]. These authors reported the use of different needle 293 sizes and bevel angles (22G and 20G, 20º and 30º respectively). In the current study, as 294 previously described, a blunt bevel (30º) was used.” - I suggest placing this description after the ultrasound description - that is, in line 270.

A: This part of the text describes important features that could explain the distribution pattern, which is discussed in the two previous paragraphs. We consider that placing the mentioned paragraph in the US section would made it difficult to understand. 

Q: Line 309 - Please indicate the source for the statement “This locoregional block has shown inhalant anaesthetic-sparing properties.”

A: This reference (Touzot-Jourde) has been clarified in the text adding the reference at the end of that phrase.

Q: Lines 311-315 - I agree. Here it is stated that it can only desensitize the cranial part, but the rest of the document mentions the cranial and middle abdominal wall. Please clarify.

A: We have changed all the statements following the same line we have mentioned in the answer to the commentary about lines 235-237. All the changes are highlighted.

Q: Lines 328-329 - I suggest removing the sentence “On the same line, a more recently described ultrasound-guided block is the QL [41,42]” and references. They do not add anything to the work.

A: QL, TAP and RSB are nowadays the three more common alternatives used to provide an ultrasound-guided analgesia to the abdomen. However, as has been described that the desensitized area obtained with this three blocks can be different, so we consider important to discuss the different alternatives to give to the reader a more complete information.

Q: Line 333 - Please remove the author's name and include just the number according to the journal's formatting rules.

A: The name has been removed

Conclusion:

Q: Clarify the issue of the middle abdomen

A: Clarified.

Reviewer 2 Report

Comments and Suggestions for Authors

Dear authors,

Thank you for submitting your article to Animals. Below are some comments that I hope will help in improving the manuscript.

Best wishes.

Summary/Abstract:

Line 17: I would change “at that level” to “within the rectus sheath.”

Line 23 and 68: As the RSB has already been employed in clinical trials, I would say the block feasibility has already been proven. I suggest limiting the objective/hypothesis to assessing the injectate distribution.

Lines 23, 35, 236: As T10 and T13 were stained only in 60% and 50% of cases, respectively, I would not talk about “consistent staining.” This technique, with the volume employed, consistently stained T11 and T12.

Lines 39 (and also in the discussion and conclusion): As T13 was stained in 50% of cases, I would not say that this technique may be used to provide analgesia in the middle abdominal region. Since T13 innervates the dermatome caudal to the umbilicus (Otero et al. 2021; Garbin et al. 2022), the technique may desensitize the region of the abdomen cranial to the umbilicus. Similarly, I would limit the applicability of the technique to procedures with an incision cranial to the umbilicus.

Introduction:

General: I suggest using veterinary references as they are available. For example, book chapters on fascial plane blocks and RS blocks have been published:

https://doi.org/10.1002/9781119514183.ch12

https://doi.org/10.1002/9781119514183.ch20

Line 47: The RS and TAP are considered “superficial fascial plane.”

Line 59: Other cadaveric studies on RS block have been performed in calves (Ferreira et al. 2022) and pigs (Ienello et al. 2022). Moreover, your reference 24 is a cadaveric study in horses, which should be reported here rather than in line 64.

Line 65: The sentence is unclear. Do the authors mean that the clinical studies showed dermatome coverage compatible with cadaveric studies? If so, this does not apply to cats, as there have been no anatomical studies so far. If, instead, the authors refer to the block analgesic efficacy, it is debatable as data in veterinary medicine are limited. For example, in an investigation in calves (n=7/group), the analgesic effects were poor compared to the control group. In horses, the data are limited to mechanical nociceptive threshold measurements (n=6/group), etc.

Lines 59-65: Cranial, middle and caudal abdomen might need to be clarified. If possible, I would report the thoracolumbar nerves involved.

Materials and Methods:

Line 74: Consider “17 cats were assessed for eligibility”.

Line 81: based on the inclusion criteria, four cats used in your study should be excluded. Maybe the authors meant “with a BCS 3/9 and 7/9.”

Lines 96-97, 124, 138 (etc.): I would indicate only the operators' initials.

Line 110: Please explain why you used such a long and big needle. Usually, 21-22G 40-60 mm needles are recommended for small patients.

https://doi.org/10.1002/9781119514183.ch20

Please expand it in the discussion section.

Line 118 and lines 262-264: If the needle was introduced where the body of the TA disappears, the explanation you gave of using the muscle to provide further protection and avoid intraperitoneal injection does not stand. Please rephrase.

Line 120: Please change “bubble” with “hydrodissection” or “anechoic pocket.”

Line 123: Was the needle visualization assessed using a scale or registered by taking and storing US-images?

Line 126: RSB was never used throughout the manuscript.

RA can be abbreviated after “rectus abdominis muscle” in the legend. It is not necessary to repeat it.

Line 143: Please change “skinned up” with scientific terminology.

Line 147: Please add a reference for the “successful staining.” For example, Otero et al. 2021; Garbin et al. 2022.

Line 150: Paragraph format: “2.3 statistical analysis”

Results:

Line 159: delete “four cat cadavers” as it is a repetition. Start with “One male and three female cat cadavers…”

Line 162-168: You clearly explain how the muscles change position; however, I would better explain how this affects the RS because the target of your injection is the RS compartment.

Line 182: I would also explain that T12 ends cranial and T13 caudal to the umbilicus. A reader unfamiliar with cat anatomy might find it challenging to comprehend the pathway of the nerves and the innervation of the abdomen (dermatomes).

Line 183: “followed the same path than TA.” Did the authors mean T13?

Line 184: This is a comment, not a result. Move and expand it in the discussion.

Line 187: Maybe “the second and third lumbar nerves run within the RS.

Line 188: I would rephrase to say that there are individual variations on the location of L1 related to the RS. I suggest reporting how many L1 nerves were in the RS compartment (as you did 4 cadavers, you should have 8 L1 nerves, X ending within the RS and XX not).

Lines 191-193: If the cadavers were excluded before performing the block, they were only assessed for eligibility. Choose the more appropriate terminology and use it here and in the abstract.

Line 195: The BCS range should be (3-7).

Table 1: You could also add a column with the sex. Could you also report the cats' age?

Line 207: In Figure 4, the contrast spread does not reach the pelvis. Please correct the sentence or change the image.

Lines 209-212: Figure legend. Some words do not need to be abbreviated (for example, ninth thoracic and fifth lumbar vertebrae).

As you abbreviated L for lumbar spinal nerve, I suggest changing L and R to LL and RL, respectively. Please correct Figure 4A.

Line 220: Use the abbreviation throughout the manuscript (RV).

Please delete “consistently.”

Lines 227-228: Please check the format. These two lines are part of the figure legend.

Discussion:

Line 237: As mentioned previously, only T11 and T12 were consistently stained.

Lines 237-238: I would rephrase as only the portion cranial to the umbilicus will be desensitized. In fact, T13 (caudal to the umbilicus) was stained only in 50% of injections.

Lines 238-240: It is irrelevant to talk about abdominal innervation in dogs. As you stated later, the abdominal innervation of cats has already been described (Otero et al. 2021; Garbin et al. 2022). I would focus on the innervation at the level of the RA muscle or the relationship between nerves and RS.

Line 244: Please rephrase. Do the authors mean that the thoracolumbar nerves run superficial to the TA and that, in the abdominal region caudal to the umbilicus, they are located ventral to the RA rather than dorsal?

Line 246: What region?

Lines 249-258: These findings in human medicine are irrelevant as it has been shown in cats and dogs that a subcostal approach to the TAP must be added to stain/desensitize the nerves cranial to T12. Focus your discussion on small animals. I would delete or rewrite the paragraph.

Lines 262-264: See comment on line 118.

Lines 273-275: An anesthesiologist could debate whether a volume based on "ideal" weight should have been administered. Moreover, cats with a BCS of 7/9 are usually not included in the analysis as they may be considered overweight and represent a “special” population.

Even though the cranio-caudal spread was longer than in the other cats, the nerves stained were only 3 or 4 (no difference!)

Lines 275-278: I would explain your reasoning better because volume and concentration are separate things. It is possible to use high volume by diluting the anesthetic (reducing the concentration) and meeting the recommended dose of 2 mg/kg.

Out of curiosity, why did you limit your study to a single injectate volume rather than comparing two volumes? It appears that the RS cannot be used as a sole LRA technique for most surgical abdominal procedures in cats as in this species the surgical incision often extends caudal to the umbilicus. If two techniques must be combined (i.e. RSB and TAPB), a volume of 0.4 ml/kg only for the RSB cannot be used. Please expand this possibility in the discussion.

Line 280-281: If L1 is not within the RS, how can a larger volume stain this nerve? Please rephrase.

Line 282: I agree that there are no studies on bupivacaine via RS in cats; however, Garbin et al. (35) used a TAP subcostal approach (which appears similar to the RSB) combined with a lateral TAPB and found a PK similar to the intraperitoneal injection (Benito et al. 2016; DOI: 10.2460/ajvr.77.6.641). Bupivacaine via RS may have the same PK.

Please explain why you used a solution of methylene blue and iopamidole (1:1), considering that the contrast affects the injectate spread (DOI: 10.1016/j.vaa.2020.01.003 ). A more diluted solution was used in cats (Garbin et al. 2022) and dogs (https://doi.org/10.1111/jsap.12639).

Line 287: did the authors find any vertebral abnormalities in these cats? Please expand.

Lines 287-288: Please explain better why an abnormality at the vertebral level would affect the injectate distribution at the RS level. Otherwise, delete the sentence.

Lines 293-294: It needs to be clarified that you refer to the authors of previous studies. Maybe rephrase by saying that for the TAP, a needle of xx was used, while for the QL, a needle of ... However, I would only explain that you used a needle with a bevel of 30 degrees, differently from the one used by Garbin et al. and the rationale behind your choice (echogenic needle vs spinal needle; different bevel angle, etc).

Line 303: delete “consistently.”

Line 305: As previously mentioned, I would say cranial to the umbilicus (T13 stained in 50% of cases).

Lines 305-309. It may not be related to the volume used but the injection point. In dogs, the injection was performed immediately cranial to the umbilicus. Here, you injected 2 cm cranial to it, which, in my experience, is mid-way from the xiphoid and the umbilicus (precisely at the level of the T11 terminal end).

Lines 316-317: In my experience, a wide incision is required for splenectomy. I would probably state that the RS block with the volume and technique in this study may be used for a procedure with a midline incision cranial to the umbilicus (but further trials are needed).

Line 322: The quadratus lumborum was already abbreviated.

Lines 324-325: all this paragraph is irrelevant to your study. Please delete.

Conclusion:

As mentioned in my previous comments, delete “the block is feasible” and “consistently.”

The conclusion is misleading: this technique poorly stained T13. I would limit your conclusion to what was observed. As T11 and T12 were constantly stained, the portion cranial to the umbilicus may be desensitized in live cats undergoing cranial midline laparotomy.

(Maybe) Further studies are necessary to improve the technique or investigate the combination of RS block with lateral TAP block to desensitize all nerves responsible for the innervation of the abdomen.

Author Response

Summary/Abstract:

Q: Line 17: I would change “at that level” to “within the rectus sheath.”

A: Changed

Q: Line 23 and 68: As the RSB has already been employed in clinical trials, I would say the block feasibility has already been proven. I suggest limiting the objective/hypothesis to assessing the injectate distribution.

A: Changed to: “Our results showed a consistent staining of the rami ventrales of several spinal nerves (T11-T12) and frequent staining of T10 and T13”

And: “We hypothesise that this technique would cover the rami ventrales (RV) …”

Q: Lines 23, 35, 236: As T10 and T13 were stained only in 60% and 50% of cases, respectively, I would not talk about “consistent staining.” This technique, with the volume employed, consistently stained T11 and T12.

A: Changed to “and a consistent staining of the rami ventrales of several spinal nerves (T11-T12) and frequent staining of T10 and T13”

Line 35, “consistent” has been deleted.

Line 236 changed to “can consistently stain the RV of T12 and T13 and frequently stain T10 and T13”

Q: Lines 39 (and also in the discussion and conclusion): As T13 was stained in 50% of cases, I would not say that this technique may be used to provide analgesia in the middle abdominal region. Since T13 innervates the dermatome caudal to the umbilicus (Otero et al. 2021; Garbin et al. 2022), the technique may desensitize the region of the abdomen cranial to the umbilicus. Similarly, I would limit the applicability of the technique to procedures with an incision cranial to the umbilicus.

A: Thank you for your remark, this point has been a reason of debate between the authors of this article. We have changed the statement to be similar to the one you suggested.

Introduction:

Q: General: I suggest using veterinary references as they are available. For example, book chapters on fascial plane blocks and RS blocks have been published:

https://doi.org/10.1002/9781119514183.ch12

https://doi.org/10.1002/9781119514183.ch20

A: We would like to thank you for the recently published references, they represent a pertinent guide to it. We have added it as references to the text. However, the authors would like to keep Elsharkawy and Chin’s references as they complete and expand the information given in the veterinary texts.

Q: Line 47: The RS and TAP are considered “superficial fascial plane.”
A: Changed 

Q: Line 59: Other cadaveric studies on RS block have been performed in calves (Ferreira et al. 2022) and pigs (Ienello et al. 2022). Moreover, your reference 24 is a cadaveric study in horses, which should be reported here rather than in line 64.

A: Thank you for your remark. At the time of writing the first draft of the text, we had doubts whether include or not these references. After your suggestion, we have decided to include them in the text.

Q: Line 65: The sentence is unclear. Do the authors mean that the clinical studies showed dermatome coverage compatible with cadaveric studies? If so, this does not apply to cats, as there have been no anatomical studies so far. If, instead, the authors refer to the block analgesic efficacy, it is debatable as data in veterinary medicine are limited. For example, in an investigation in calves (n=7/group), the analgesic effects were poor compared to the control group. In horses, the data are limited to mechanical nociceptive threshold measurements (n=6/group), etc.

A: Thank you for the remark. Yes, the original sentence could be misleading. Additionally, we agree that the data is debatable due to the low N, further clinical studies would be needed to assess the suitability of the RS block. We have rephrased it. A discussion about the cats’ clinical studies can be found in the Discussion section. 

Q: Lines 59-65: Cranial, middle and caudal abdomen might need to be clarified. If possible, I would report the thoracolumbar nerves involved.

A: The different anatomic morphologies of these species made it difficult to unify the thoracolumbar nerves of several species. The authors decided to divide the abdomen in three regions as described in the “Illustrated Veterinary Anatomical Nomenclature” (ISBN 978-3-13-242517-0) 4th edition, p.6-7. This reference has been added.

Materials and Methods:

Q: Line 74: Consider “17 cats were assessed for eligibility”.

A: Changed

Q: Line 81: based on the inclusion criteria, four cats used in your study should be excluded. Maybe the authors meant “with a BCS ≥ 3/9 and ≤ 7/9.”

A: The phrase has been changed to: Cadavers were enrolled in this study if their body condition score (BCS) was superior to 2/9 and inferior to 8/9 according to the WSAVA Classification.

Q: Lines 96-97, 124, 138 (etc.): I would indicate only the operators' initials.

A: Changed

Q: Line 110: Please explain why you used such a long and big needle. Usually, 21-22G 40-60 mm needles are recommended for small patients.

https://doi.org/10.1002/9781119514183.ch20

Please expand it in the discussion section.

A: Expanded in the discussion section.

Q: Line 118 and lines 262-264: If the needle was introduced where the body of the TA disappears, the explanation you gave of using the muscle to provide further protection and avoid intraperitoneal injection does not stand. Please rephrase.

A: Thank you for your suggestion. As this important part of the text remained unclear, we have decided to rephrase it and to change the Figure 2 (following as well some of the suggestions of other reviewer)

The text have change to:

  • Once the external sheath of the RA was pierced, the needle was reoriented and advanced through the RA until being in contact with the RS at the medial edge of the belly of TA
  • The pressure exerted to pierce this external sheath increases the risk of perforating the entire thickness of RA, causing an accidental intraperitoneal or visceral puncture. […] The presence of the OIA aponeurosis (Fig 3) in this location ensures injection within the RS.

Q: Line 120: Please change “bubble” with “hydrodissection” or “anechoic pocket.”

A: Changed to “anechoic pocket”

Q: Line 123: Was the needle visualization assessed using a scale or registered by taking and storing US-images?

A: The US-images were stored, but we did not consider using a scale as the visualization was optimal in every injection. Despite this, all the images were reassessed afterwards to reconfirm the needle localization and visualization. 

Q: Line 126: RSB was never used throughout the manuscript.

RA can be abbreviated after “rectus abdominis muscle” in the legend. It is not necessary to repeat it.

A: All these proposed changes have been made.

Q: Line 143: Please change “skinned up” with scientific terminology.

A: The phrase has been changed to: A ventral midline incision was carried out and the skin was dissected up to the dorsal region to expose the underlying tissues

Q: Line 147: Please add a reference for the “successful staining.” For example, Otero et al. 2021; Garbin et al. 2022.

A: The suggested references have been added to the text.

Q: Line 150: Paragraph format: “2.3 statistical analysis”

A: Changed

Results:

Q: Line 159: delete “four cat cadavers” as it is a repetition. Start with “One male and three female cat cadavers…”

A: Changed

Q: Line 162-168: You clearly explain how the muscles change position; however, I would better explain how this affects the RS because the target of your injection is the RS compartment.

A: This paragraph has been added to the text: “Thus, in the cranial abdomen, the RS is formed by the OIA and TA aponeurosis. In the middle abdomen, it is formed only by the TA aponeurosis. It ends in the aforementioned change of TA position in the caudal region.”

Q: Line 182: I would also explain that T12 ends cranial and T13 caudal to the umbilicus. A reader unfamiliar with cat anatomy might find it challenging to comprehend the pathway of the nerves and the innervation of the abdomen (dermatomes).

A: These underlined changes have been made to the text:

“The tenth (T10), eleventh (T11) and T12 RV were found emerging from the costal arch and advancing between the OIA aponeurosis and the TA, towards the RA muscle where they ended, cranial to the umbilicus. Finally, at the lateral aspect of the abdominal wall the RV of T13, L1, L2 and L3 were also found lying between the OIA and TA muscles. At this point RV of T13 continued until it ended in the RA caudal to the umbilicus.”

Q: Line 183: “followed the same path than TA.” Did the authors mean T13?

A: No, we meant that L1 travelled together with the TA muscle on the TA plane.

Q: Line 184: This is a comment, not a result. Move and expand it in the discussion.

A: Deleted from the results section.

Q: Line 187: Maybe “the second and third lumbar nerves run within the RS.

A: It is not possible, as the RS is not present in that abdominal region.

Q: Line 188: I would rephrase to say that there are individual variations on the location of L1 related to the RS. I suggest reporting how many L1 nerves were in the RS compartment (as you did 4 cadavers, you should have 8 L1 nerves, X ending within the RS and XX not).

A: Unfortunately, due to the small size of the nerve and the particularity of the anatomical region where L1 pass through (where TA changes from a dorsal to a ventral position respectively to RS), determining whether a particular L1 was in the RS compartment or not was ambiguous and challenging. In order not to make a scientific inaccurate statement, we couldn’t affirm nor deny its involvement into the RS. In addition, staining of L1 nerve could be due to caudal communication between the caudal region of the RS and the transversus abdominis plane. This hypothesis could explain the anecdotical L1 staining.

Q: Lines 191-193: If the cadavers were excluded before performing the block, they were only assessed for eligibility. Choose the more appropriate terminology and use it here and in the abstract.

A: The terminology has been changed. 

Q: Line 195: The BCS range should be (3-7).

A: Corrected

Q: Table 1: You could also add a column with the sex. Could you also report the cats' age?

A: Cats’ age cannot be reported as these data were not declared at the moment of body donation in many cases. Due to the lack of this information and given that another reviewer has suggested deleting this table because all the relevant information has already been explained, we have decided to delete it.

Q: Line 207: In Figure 4, the contrast spread does not reach the pelvis. Please correct the sentence or change the image.

A: It did reach the pelvis in one cat, which is not the one in B. We have relocated the reference to the figure at the beginning of the paragraph.

Q: Lines 209-212: Figure legend. Some words do not need to be abbreviated (for example, ninth thoracic and fifth lumbar vertebrae).

A: These suggestions have been added to the figure legend

Q: As you abbreviated L for lumbar spinal nerve, I suggest changing L and R to LL and RL, respectively. Please correct Figure 4A.

A: The figure has been corrected.

Q: Line 220: Use the abbreviation throughout the manuscript (RV).

A: All the possible abbreviations have been changed. We have decided not to abbreviate “rami ventrales” from the figure legends, simple summary and abstract to make their lecture easier as isolated parts of the text.

Q: Please delete “consistently.”

A: Deleted

Q: Lines 227-228: Please check the format. These two lines are part of the figure legend.

A: These two lines have been corrected and included into the figure legend.

Discussion:

Q: Line 237: As mentioned previously, only T11 and T12 were consistently stained.

A: This line has been change to: “The results of the present study show that an ultrasound-guided injection of 0.4 mL kg-1 of a mixture of methylene blue and iopromide can consistently stain the RV of T12 and T13 and frequently stain T10 and T13”

Q: Lines 237-238: I would rephrase as only the portion cranial to the umbilicus will be desensitized. In fact, T13 (caudal to the umbilicus) was stained only in 50% of injections.

A: The statement has been rephrased

Q: Lines 238-240: It is irrelevant to talk about abdominal innervation in dogs. As you stated later, the abdominal innervation of cats has already been described (Otero et al. 2021; Garbin et al. 2022). I would focus on the innervation at the level of the RA muscle or the relationship between nerves and RS.

A: This has been rephrased to: “Our anatomical findings in cat cadavers showed that the ventral abdominal wall of cats is innervated by the RV of the spinal nerves from T10 to L3 [4,5,16]”

Q: Line 244: Please rephrase. Do the authors mean that the thoracolumbar nerves run superficial to the TA and that, in the abdominal region caudal to the umbilicus, they are located ventral to the RA rather than dorsal?

A: These observations are explained in the results sections (lines 191-193). In the discussion, we have decided to focus on the nerves' involvement into the RS.

Q: Line 246: What region?

A: This part of the text has been rephrased to: “However, the RV of L2 and L3 run caudal to the change of TA position, where RS is discontinued.”

Q: Lines 249-258: These findings in human medicine are irrelevant as it has been shown in cats and dogs that a subcostal approach to the TAP must be added to stain/desensitize the nerves cranial to T12. Focus your discussion on small animals. I would delete or rewrite the paragraph.

A: The authors think that is important to state that TAP block was not able to desensitize the midline in human volunteers, but lateral to it. These kinds of studies have not been performed in veterinary medicine and we find it important because it represents one of the reasons why Rectus Sheath block could be performed instead of TAP block. Despite the study of Garbin 2023, data about the advantages of one anesthetic technique over the other are still lacking.

Q: Lines 262-264: See comment on line 118.

A: The phrase has been changed to: “The pressure exerted to pierce this external sheath increases the risk of perforating the entire thickness of RA, causing an accidental intraperitoneal or visceral puncture.”

Q: Lines 273-275: An anesthesiologist could debate whether a volume based on "ideal" weight should have been administered. Moreover, cats with a BCS of 7/9 are usually not included in the analysis as they may be considered overweight and represent a “special” population.

Even though the cranio-caudal spread was longer than in the other cats, the nerves stained were only 3 or 4 (no difference!)

A: Thank you for the remark, we agree.

Q: Lines 275-278: I would explain your reasoning better because volume and concentration are separate things. It is possible to use high volume by diluting the anesthetic (reducing the concentration) and meeting the recommended dose of 2 mg/kg.

A: We have added the next section to the text: “A further dilution of the local anaesthetic (bupivacaine 0.125% or ropivacaine 0.125%) could increase the total volume without exceeding the maximum recommended dose. However, a reduction in the concentration of the drug could reduce its effectiveness ”
Adding this reference: https://doi.org/10.1002/9781119421375.ch17

Q: Out of curiosity, why did you limit your study to a single injectate volume rather than comparing two volumes? It appears that the RS cannot be used as a sole LRA technique for most surgical abdominal procedures in cats as in this species the surgical incision often extends caudal to the umbilicus. If two techniques must be combined (i.e. RSB and TAPB), a volume of 0.4 ml/kg only for the RSB cannot be used. Please expand this possibility in the discussion.

A: At the time of the study design process, no clinical or cadaveric studies had been published, so we decided to assess the feasibility of the technique in cats at a fixed volume to increase the number of hemiabdomens included. As you pointed out, further studies should investigate the differences in volume, injection point or combination of several techniques.  As stated in the study, this technique could be employed in upper abdominal surgery, such as gastrotomies.

These new sentences have been added: “In addition, combinations of two or more of the aforementioned techniques could enhance anaesthesia properties and the area of desensitasion. Further clinical studies are needed to assess which locoregional technique or combination would provide a better analgesia as well as anaesthetic and opioid-sparing effects.”

Q: Line 280-281: If L1 is not within the RS, how can a larger volume stain this nerve? Please rephrase.

A: The integration of L1 into the RS remains unclear, further studies should focus on this specific aspect of the anatomy of the cats. Communication between the caudal part of the RS and the TA plane could also explain L1 staining. This hypothesis has not been proved. 

Q: Line 282: I agree that there are no studies on bupivacaine via RS in cats; however, Garbin et al. (35) used a TAP subcostal approach (which appears similar to the RSB) combined with a lateral TAPB and found a PK similar to the intraperitoneal injection (Benito et al. 2016; DOI: 10.2460/ajvr.77.6.641). Bupivacaine via RS may have the same PK.

A: We agree that it could be similar, unfortunately no studies have confirmed this hypothesis in RS block. Using this data from TAP block  as a statement for the RS block could be considered scientifically inaccurate.

Q: Please explain why you used a solution of methylene blue and iopamidole (1:1), considering that the contrast affects the injectate spread (DOI: 10.1016/j.vaa.2020.01.003 ). A more diluted solution was used in cats (Garbin et al. 2022) and dogs (https://doi.org/10.1111/jsap.12639).

A: We used this solution to make the results comparable to other studies from the same research group (10.3390/ani13243798,  10.3390/ani12010018 and 10.3390/ani13132214) as well as other research groups (10.3390/ani11102945). However, it is acknowledged in the limitations section that the election of these molecules and solutions could alter the results. Unfortunately, to the authors' knowledge, there are no guidelines regarding this issue and any mixture employed could theoretically alter the results.

However, we will consider your suggestion in the study design of future studies.

These new phrases have been added: “The mixture (1:1) employed was chosen to make the results comparable to other studies and block techniques in dogs [33–35] and cats [9]. Other solution concentrations have been studied in cats [6], showing that lower contrast concentrations could be employed. In addition, one study in dogs showed differences in spread extension depending on the contrast concentration employed [36].” The references suggested have been added. 

The consequences of this election are stated in the limitations section.

Q: Line 287: did the authors find any vertebral abnormalities in these cats? Please expand.

A: We did not found any. We gave decided to delete the sentence and reference.

Q: Lines 287-288: Please explain better why an abnormality at the vertebral level would affect the injectate distribution at the RS level. Otherwise, delete the sentence.

A: Due to the lack of these abnormalities in our study we have decided to delete the sentence and the reference

Q: Lines 293-294: It needs to be clarified that you refer to the authors of previous studies. Maybe rephrase by saying that for the TAP, a needle of xx was used, while for the QL, a needle of ... However, I would only explain that you used a needle with a bevel of 30 degrees, differently from the one used by Garbin et al. and the rationale behind your choice (echogenic needle vs spinal needle; different bevel angle, etc).

A: The sentences have been rewritten:  “These authors reported the use of different needle sizes and bevel angles (22G, 20º for the TAP block study and 20G, 30º for the QL).”

A new phrase has been added: “The type of needle employed could also be a differentiating factor. A sonovisible needle was chosen in this study to enhance its visualization throughout its path.”

The chapter 3 of this book has been added as reference: ISBN 978-950-555-465-2

Q: Line 303: delete “consistently.”

A: Deleted

Q: Line 305: As previously mentioned, I would say cranial to the umbilicus (T13 stained in 50% of cases).

A: The sentence has been rephrased to: “In a clinical scenario, this technique could potentially provide anaesthesia cranial to the umbilicus and, potentially, the middle abdominal midline”

Q: Lines 305-309. It may not be related to the volume used but the injection point. In dogs, the injection was performed immediately cranial to the umbilicus. Here, you injected 2 cm cranial to it, which, in my experience, is mid-way from the xiphoid and the umbilicus (precisely at the level of the T11 terminal end).

A: We greatly appreciate your suggestion; we have rephrased the text: Compared to a previous research conducted on cadaver Beagle dogs using a higher volume of injectate (0.5 mL kg-1) and a different injection point (at the level of the umbilicus) [21], we had the same nerves stained in our study. However, the percentage of nerve staining was slightly different, showing our results a more consistent cranial distribution of the dye potentially due to the more cranial injection point.

Q: Lines 316-317: In my experience, a wide incision is required for splenectomy. I would probably state that the RS block with the volume and technique in this study may be used for a procedure with a midline incision cranial to the umbilicus (but further trials are needed).

A: We have rephrased this statement: “However, it may be appropriate for procedure with a midline incision cranial to the umbilicus” 

Q: Line 322: The quadratus lumborum was already abbreviated.

 A: We have changed the sentence to: “(epidural, QL or caudal paravertebral blocks)”

Q: Lines 324-325: all this paragraph is irrelevant to your study. Please delete.

A: We appreciate your suggestion, however, as part of the discussion of the subject, we consider important to emphasize the alternatives to the RS block as well as its advantages and disadvantages, so an unfamiliar reader can acquire a quick and overall view of the current knowledge in abdominal anaesthetic techniques.

Conclusion:

Q: As mentioned in my previous comments, delete “the block is feasible” and “consistently.”

A: We have rephrased the statement

Q: The conclusion is misleading: this technique poorly stained T13. I would limit your conclusion to what was observed. As T11 and T12 were constantly stained, the portion cranial to the umbilicus may be desensitized in live cats undergoing cranial midline laparotomy.

A: The conclusion has been rephrased, as well as the simple summary and the abstract.

Q: (Maybe) Further studies are necessary to improve the technique or investigate the combination of RS block with lateral TAP block to desensitize all nerves responsible for the innervation of the abdomen.

A: We agree.

Round 2

Reviewer 1 Report

Comments and Suggestions for Authors

Thank you for the corrections and responses. The manuscript is significantly better in this version.